# Endothelial Progenitor Cells and Rheumatoid Arthritis: Response to Endothelial Dysfunction and Clinical Evidences

**DOI:** 10.3390/ijms222413675

**Published:** 2021-12-20

**Authors:** Klara Komici, Angelica Perna, Aldo Rocca, Leonardo Bencivenga, Giuseppe Rengo, Germano Guerra

**Affiliations:** 1Department of Medicine and Health Sciences, University of Molise, 86100 Campobasso, Italy; angelica.perna@unimol.it (A.P.); aldo.rocca@unimol.it (A.R.); germano.guerra@unimol.it (G.G.); 2Department of Advanced Biomedical Sciences, University of Naples “Federico II”, 80131 Naples, Italy; leonardobencivenga@gmail.com; 3Gérontopôle de Toulouse, Institut du Vieillissement, CHU de Toulouse, 310000 Toulouse, France; 4Department of Translational Medical Sciences, University of Naples “Federico II”, 80131 Naples, Italy; giuseppe.rengo@unina.it; 5Istituti Clinici Scientifici Maugeri SpA Società Benefit (ICS Maugeri SpA SB), 82037 Telese Terme, Italy

**Keywords:** rheumatoid arthritis, endothelial dysfunction, endothelial progenitor cells, endothelial colony forming cells, myeloid angiogenic cells

## Abstract

Rheumatoid Arthritis (RA) is a chronic autoimmune inflammatory disease characterized by the swelling of multiple joints, pain and stiffness, and accelerated atherosclerosis. Sustained immune response and chronic inflammation, which characterize RA, may induce endothelial activation, damage and dysfunction. An equilibrium between endothelial damage and repair, together with the preservation of endothelial integrity, is of crucial importance for the homeostasis of endothelium. Endothelial Progenitor Cells (EPCs) represent a heterogenous cell population, characterized by the ability to differentiate into mature endothelial cells (ECs), which contribute to vascular homeostasis, neovascularization and endothelial repair. A modification of the number and function of EPCs has been described in numerous chronic inflammatory and auto-immune conditions; however, reports that focus on the number and functions of EPCs in RA are characterized by conflicting results, and discrepancies exist among different studies. In the present review, the authors describe EPCs’ role and response to RA-related endothelial modification, with the aim of illustrating current evidence regarding the level of EPCs and their function in this disease, to summarize EPCs’ role as a biomarker in cardiovascular comorbidities related to RA, and finally, to discuss the modulation of EPCs secondary to RA therapy.

## 1. Introduction

Rheumatoid Arthritis (RA) is a chronic inflammatory joint disease, with a prevalence of about 1% worldwide [1]. The peak incidence is registered among subjects of 60 years of age, and women are characterized by a higher risk of developing RA [2]. Pain, swelling and stiffness of feet and hands joints represent the classic presentation of the diseases. However, non-specific symptoms such as fatigue and fever may appear before classic symptoms. Extra-articular manifestations and complications of RA include: nodules, rheumatoid vasculitis, osteoporosis, vertebral fractures, progressive disability and a lower quality of life. Compared to healthy controls, RA patients may have lower bone quality and RA is considered a risk factor for vertebral fractures [3,4]. Patients with severe RA disease have a 6.5-fold higher risk for the development of vertebral fractures compared to age-matched controls, and the current use of steroids and disease-modifying anti-rheumatic drugs (DMARDs) are inversely related to vertebral fractures [4]. Interstitial lung disease, neurological disorders, atherosclerosis and cardiovascular disease are common comorbidities associated with RA [5]. While interstitial lung disease may both constitute a manifestation of RA or a complication of RA therapies [6], treatment with targeted biologic factors have been demonstrated to reduce cardiovascular risk [7]. 

The risk factors for RA are relatively unknown, however, a combination of genetic and environmental factors, including smoking, viral infections, the gastro-intestinal microbiome and periodontitis have been associated with the development of RA [1]. These factors are promotors of the inflammatory response and induce a complex autoimmune process, characterized by the presence of anticitrullinated protein antibodies, antibodies to rheumatoid factor (RF), nuclear antigens and auto-antigens that cross-talk with bacterial or viral antigens. The onset of RA is proceeded by a preclinical RA stage, wherein genetic factors such as shared-epitope-positive HLA-DRB1 alleles, PTPN22 variant and environmental stressors promote post-translational protein modifications such as citrullination or carbamylation, and generate neo-epitopes of autologous proteins such as fibrinogen, fibronectin, collagen, and vimentin. This process results in the loss of self-tolerance and the development of autoantibodies against anticitrullinated protein antibodies, antibodies directed against the Fc portion of immunoglobulins, and RF [8,9,10]. Consequently, the immune response results in the activation and differentiation of T cells, release of IL-2, Il-6, Il-17, Il-21 and INF-gamma and the activation of B-cells, which secret autoantibodies. An abnormal differentiation of naive CD4^+^ T cells into highly proliferative and proinflammatory effector cells, leads to tissue tolerance failure and early synovitis [11]. The defective transition of the T- and B- cell population from naive to effector and memory states has been suggested as the principal mechanism involved in the development of tissue tolerance [12,13]. Of note, in experimental models of arthritis, in vitro generated collagen-II specific B cells induced immune tolerance [14]. Additionally, Il-17, INF- γ and immune complexes activate macrophages, which release IL-1, IL-6 and TNF and activate fibroblasts. Fibroblast activation and proliferation may adopt pro-inflammatory and tissue invasive functions by releasing metalloproteinases (MMPs) and causing the differentiation of macrophages to osteoclasts. In addition, the release of IL-11 by activated fibroblasts controls fibroblasts’ trafficking and production of IL-8 and VEGF, which results in angiogenesis and potentiates neovascularization [15]. The evasion of the synovial membrane by macrophages and fibroblasts leads to cartilage degradation and bone erosion. A sustained immune response, and chronic inflammation, which characterize RA, may induce endothelial activation, damage and dysfunction. Furthermore, the epithelial–mesenchymal transition has been associated with RA, and IL-23 seems to intermediate the transition of alveolar epithelial cells (ATI) to a mesenchymal phenotype through mTOR/S6 signaling [16]. ATII cells have an important role in the regulation of innate immunity and inflammatory mediators such as transforming growth factor β, which is increased in RA, \nd may modify the ATII secretory profile, which in turn may directly activate fibroblasts migration and proliferation, resulting in lung tissue remodeling and fibrosis development [17,18]. An equilibrium between endothelial damage and repair, together with the preservation of endothelial integrity is of crucial importance for the homeostasis of the endothelium. Endothelial Progenitor Cells (EPCs) represent a heterogenous cell population, characterized by the ability to differentiate into mature ECs, which contribute to vascular homeostasis, neovascularization and endothelial repair. Modification in number and in the function of EPCs has been described in physiological aging, in subjects with high cardiovascular risk, and in several chronic inflammatory and auto-immune conditions, such as systemic lupus erythematosus [19,20,21,22]. Reports focused on the number and function of EPCs in RA are characterized by conflicting results and discrepancies that exist among different studies. However, given the crucial connection between EPCs, endothelial damage and chronic inflammation, a modification of EPC function should be present in RA. In this review, we describe EPCs’ response to endothelial modification in RA, with the aim to illustrate current evidences regarding EPCs’ level and function in RA, summarize EPCs’ role as a biomarker in cardiovascular comorbidities related to RA and finally, to discuss the modulation of EPCs secondary to RA therapy. 

## 2. EPCs Classification and Their Role in Endothelial Repair

EPCs are broadly defined as a wide heterogeneous group of cells that are able to differentiate into ECs and contribute to the formation of new blood vessels [23]. Despite the significant progress made in this field, it is still difficult to standardize the identification of EPCs and to correctly define their characteristics and properties. However, based on their biological functions and phenotype, two distinct subtypes of EPCs have been proposed, namely, myeloid angiogenic cells (MACs) and endothelial colony forming cells (ECFCs) [24]. MACs are defined as cultured cells that are derived from peripheral blood mononuclear cells grown under endothelial cell culture conditions, which are characterized by the following surface cell markers: CD45^+^, CD14^+^, CD31^+^, and CD146^−^, CD133^−^, and Tie2^−^. MACs do not have the capacity to differentiate to ECs, but promote angiogenesis without its incorporation into the vascular lumen through a paracrine mechanism, based on the activation of IL-8/VEGFR2/ERK signaling pathways, which results in endothelial proliferation, migration and tube formation [25]. ECFCs derive from umbilical cord blood or peripheral blood mononuclear cells and are characterized by CD31+, VE-Cadherin+, von Willebrand factor^+^, CD146^+^, VEGFR2^+^, and CD45^−^ and CD14^−^ immunophenotype. Functionally, they exhibit a proliferative capacity, intrinsic in vitro and in vivo tube-forming capabilities, vascular network repair competencies and de novo blood vessels formation capacity. Their vasculogenic properties are also linked to their role as paracrine mediators via platelet-derived growth factor BB (PDGF-BB)/platelet-derived growth factor receptor (PDGFR)-β signaling [26]. Initially, the release of inflammatory cytokines and angiogenic factor, from the damaged endothelium wall, induces MAC and ECFC recruitment. MACs paracrine signaling mechanisms further enhance the migration of circulating or vascular wall ECFCs to the injury area, where they proliferate, differentiate into mature ECs and restore the endothelial integrity of the vascular wall [27].

## 3. Endothelial Dysfunction in RA

An impairment of endothelial functions has been established as a key element in the development of the atherosclerosis process and is recognized as an important factor related to cardiovascular risk in RA [28,29]. Likewise, altered endothelial reactivity has been documented in RA patients without cardiovascular risk factors and prior to atherosclerotic plaque detection, suggesting that endothelial impairment is related to a specific RA-associated chronic inflammatory condition [30]. 

Normally functioning endothelium provides a separation between blood and tissues, regulates blood flow, allows small molecules transport, and responds to signals implicated in the inflammatory process [31]. Endothelial activation typically occurs in response to inflammatory stimuli, and is characterized by an upregulation of the expression of intracellular adhesion molecule-1 (ICAM-1), vascular cell adhesion molecule-1 (VCAM-1) and E-selectin in endothelial cells (ECs). Both VCAM-1 and ICAM-1 are important in leucocyte trafficking and their increased expression is associated with a number of chronic inflammatory diseases, including RA, and the selective recruitment of EPCs to inflamed joint tissue as mediated by VCAM-1 has been reported [32]. In addition, serum concentrations of soluble ICAM-1, VCAM-1, correlated with markers of RA activity such as the erythrocyte sedimentation rate and C reactive protein levels [33]. 

Endothelial dysfunction is usually defined as a defective synthesis and release of endothelium-derived nitric oxide (NO), which promotes blood-vessel dilation [34]. Endothelial damage is characterized by the binding of immune complexes and autoantibodies to the endothelium, which leads to increased anti-fibrinolytic activity, excessive coagulation, vasoconstriction and atherosclerosis development.

Endothelial nitric oxide synthase (eNOS), regulates NO synthesis through the conversion of L-arginine to NO in the presence of tetrahydrobiopterin (BH4). The production of NO by endothelial cells inhibits leukocytes and the plate’s adhesion to the endothelium, and induces vasorelaxation. The activation of the inflammatory response and the production of IL-1, IL-6, and TNF-α is an important mechanism which implicates NO bioavailability. It has been reported that TNF-α blocks the activation of eNOS through the degradation of eNOS mRNA and by interfering with the phosphorylation of protein kinase Akt [35]. Other mechanisms implicated in NO bioavailability reduction are: L-arginine intracellular deficiency, decreased availability of BH4 co-factor, accumulation of the endogenous eNOS inhibitor dimethylarginine, and inactivation of NO through excessive generation of superoxide anion [36,37]. 

Up-regulation of L-arginase pathway and BH4 deficit trigger the transfer of an electron from NADPH to O2, mediated by eNOS, which results in superoxide anion production and, subsequently, scavenges NO to produce peroxynitrite. Experimental models have shown that ECs’ BH4-dependent eNOS regulation plays a pivotal role in maintaining vascular homeostasis [38].The reduction of NO bioavailability with the increased production of superoxide anions and peroxynitrite is defined as eNOS un-coupling. Of interest, the administration of peroxynitrite in in vitro culture of EPCs increased apoptosis and necrosis, although treatment with peroxynitrite scavenger reversed the injury [39] and the in human EPC biosynthesis of BH4 via the PTEN-AKT signaling pathway enhanced the regenerative function of EPCs [40]. 

NAD(P)H oxidase, has also been described as another source of superoxidase anion production. Indeed, an experimental model of RA revealed that NADPH oxidase is responsible for increased endothelial oxidative stress, and an in vitro administration of diphenylene iodonium chloride, an inhibitor of NAD(P)H oxidase activity, reduced the production of superoxide anions [41]. Furthermore, recent findings report that NOX4-type NADPH oxidase is important for proliferation, migration and apoptosis of EPCs [42], while a positive correlation between the NOX-mediated oxidative stress and the dysfunctions of circulating EPCs in dyslipidemia has been described, suggesting that a suppression of NOX might offer a novel strategy through which to improve EPC functions [42,43].

A contributing role of Angiotensin II (Ang-II) in ED, related to RA, has been suggested since Ang-II enhances superoxide anion production by stimulation of NADP(H) oxidase, and the treatment with angiotensin II receptor blockers (ARBs) led to a reduction of superoxide anions and the improvement of endothelial function [44,45]. an inhibition of Ang II signaling by ARBs and ACE-inhibitors has been reported to increase the number of EPCs [46]. 

## 4. EPC Response to Endothelial Dysfunction in RA

Maintaining adequate levels and function of EPCs is of particular importance for the preservation of endothelial function and is protective against the atherosclerosis process, as the chronic inflammatory condition which characterizes RA jeopardizes the endothelial integrity. Initially, the release of inflammatory cytokines and pro-angiogenic mediators is expected to trigger the mobilization and recruitment of circulating EPCs. In a collagen-induced arthritis model, EPC markers significantly increased in the peripheral blood and accumulated in inflamed synovial pannus, showing a significant increase approximately 3 weeks after disease onset [32]. Furthermore, the selective recruitment of EPCs to inflamed joint tissue is mediated by VCAM-1/the very late activation antigen 4, which provides EPC adhesion to cultured RA fibroblasts and to synovial tissue [32]. 

Experimental models revealed that VEGF plays a crucial role during the early stage of RA development, affecting neovascularization and the progression of synovitis [47]. In RA, the combination of local hypoxic conditions and the release of inflammatory cytokines such as TNF, IL-1, IL-6 and IL-18 activates macrophages and synovial tissue fibroblasts to secrete VEGF and fibroblasts growth factor. In turn, they activate ECs, induce the production of pro-proteolytic enzymes and basement membrane degradation by MMPs, which results in ECs migration and their proliferation to vascular tubules, and, lastly, pericytes are incorporated into the newly formed basement membrane [48]. The excessive expression of VEGF in RA-inflamed synovial tissue has been broadly reported on [49] and the double labeling of endothelium and pericytes/smooth muscle mural cells of synovial arthroscopic biopsies from RA, revealed that immature vessels were present from the earliest phases of RA, and their density increased in patients with a longer disease duration [50]. Of interest, a recent study observed that pericyte-derived fibroblasts contribute to fibroblast proliferation and fibrosis expansion in arthritis [51]. However, the migratory response to VEGF stimulation and EPC adherence capacity isolated from RA patients with low disease activity, was found to be reduced [47]. In this study, the overall EPC number correlated with endothelial dysfunction characterizing RA. The presence of steady, low-grade inflammation and the intake of disease-modifying antirheumatic drugs were suggested as possible explanation for the modification of the EPC number [52]. It should be mentioned that the reduced number of EPCs may also be explained by their recruitment to the injury site and the homing of EPCs to the synovial tissue might be enhanced during periods of synovitis exacerbations, resulting in reduction of circulating EPCs. Indeed, the presence of EPCs was revealed in the synovial tissue of RA patients [53], and data from an experimental and clinical model demonstrated that resistin promotes EPC homing into the synovium during RA angiogenesis via VEGF signaling [54]. In addition, CXCL16 chemokin and its receptor CXCR6 have been proposed as mediators of EPC recruitment and neovascularization in the RA joint [55].

A recent study reported that adiponectin, an inflammatory mediator secreted by adipose tissue, stimulates EPC migration and tube formation through activation of VEGF expression facilitated by MEK/ERK signaling. The inhibition of adiponectin resulted in the reduction of joint swelling, bone destruction, and angiogenic-marker expression in collagen-induced arthritis mice [56]. Another study documented that another adipokine apelin facilitated Ang1-dependent EPCs angiogenesis by inhibiting miR-525-5p synthesis via phospholipase C gamma and alpha signaling [57]. Cysteine-rich 61 proinflammatory cytokin has been shown to promote VEGF expression and to increase EPC-mediated angiogenesis in RA [58]. Although the role of sphingosine kinase 1 (S1P) in RA angiogenesis is unclear, a conditioned medium from S1P-treated osteoblasts significantly increased EPC migration and tube formation. In addition, S1P and VEGF levels were higher in synovial fluid from RA, compared to osteoarthritis patients, and infections with SphK1 shRNA reduced angiogenesis, articular swelling and cartilage erosion in the ankle joints of mice with collagen-induced arthritis [59]. TNF-*α* has also been suggested as a suppressor of EPC proliferation and migration, as corticosteroid treatment increased EPC numbers [53]. 

The impairment of NO/eNOS signaling and endothelial dysfunction has been shown to effect EPC mobilization [60]. Furthermore, endothelial dysfunction in patients with RA was associated with a reduced number and dysfunction of EPC [52].

The main mechanisms that are responsible for the activation and modification of EPCs are highlighted in Figure 1. 

Activation of ROS, NO/eNOS signaling, TNF, and other factors as adipokines, sphingosine 1 phosphate modify proliferation, recruitment and migratory properties of EPCs, which further interfere with endothelial repair/damage equilibrium and induce acceleration of atherosclerosis and other conidiations related to RA.

## 5. EPCs Levels in RA: Evidence from Clinical Studies

Different studies have compared the numbers of EPCs isolated from the peripheral blood of RA patients to healthy controls. Grisar and colleagues reported that circulating EPCs were reduced in patients affected by RA, and that active RA was associated with a depletion in EPCs numbers [61]. Furthermore, among RA patients with high serum levels of TNF-*α*, circulating EPC levels were found to be significantly decreased, while the erythropoietin level was not correlated with EPC numbers, suggesting TNF-*α* dependence and erythropoietin resistance mechanisms [53]. In line with this observation, infliximab, a chimeric antibody targeted against TNF-*α*, improved the numbers and functional properties of EPCs, in parallel with an early clinical effect, and glucocorticoid treatment also increased EPCs levels [53,62]. Another study, including low activity RA patients receiving methotrexate standard treatment, also found reduced numbers of EPCs among RA patients, and that the endothelial dysfunction among RA patients measured by forearm blood flow correlated with a reduction in overall EPCs and a reduction in their migratory properties [52]. Other evidence associated the depressed EPCs numbers in RA patients with increased asymmetric dimethyl-L-arginine levels, suggesting that oxidative stress directly up-regulates the activity of asymmetric dimethyl-L-arginine levels [63]. Even though the overall number of EPCs, compared to healthy controls, was reduced, data from this study did not reach statistical significance. The results were explained by different methods used to extract and identify EPCs, and the EPCs level was inversely correlated to RA prognostic markers such as rheumatoid factor (RF) and c reactive protein (CRP) [64]. In addition, another study associated the reduced numbers of EPCs with atherosclerosis development among RA patients [65]. In contrast, a study including patients with different disease’s activity observed increased EPC numbers. This result was explained by the different protocol used to quantify EPCs and the fact that the population was characterized by a relatively long disease duration [66]. Results from another study also failed to reflect significant differences in EPCs number by comparing early stage RA patients with healthy controls. The relatively short disease duration in this RA cohort has been suggested as an important source of discrepancies. In fact, the authors observed that EPCs population tend to decrease after diagnosis, and a significant positive correlation appeared between EPCs number and disease activity [67]. Furthermore, data from another study revealed that long-standing (more than a year) RA patients exhibited EPC depletion compared to their early stage counterparts [68]. Another study explained the contradictory results regarding EPCs number in RA patients by disease-specific factors, such as low or high expression of IFNα. From their results, EPCs were found to be significantly reduced in patients with low IFNα serum levels, whereas higher levels of IFNα were associated with a higher number of EPCs [69]. Another study, focused on RA with longer disease duration, observed a reduced number of EPCs and a correlation with angiogenic T cells (T-ang). A decrease in T-ang was detected in RA patients even at diagnosis and, at this time point, it was not related to EPCs numbers. T-ang were found to be decreased in a disease activity-dependent manner in RA patients, suggesting that specific disease and T-ang/EPCs association was partially recovered in patients with low disease activity [70]. Furthermore, it has been suggested that CD147 may play a critical role in regulating the VEGF production of activated T-ang cells by affecting Akt signaling [71]. Fenofibrate treatment did not effect EPCs expression, even though, compared to the health controls, the EPC number before treatment was lower [72], while short-term treatment of RA with TNF inhibitors was associated with increased EPCs relating a proportional decrease of disease activity [73]. 

Another study confirmed the reduction of EPCs in 126 RA patients and a reduced EPC number was associated with higher bone erosion scores in RA patients. In addition, EPC numbers were restored by anti-TNF therapy, and this increase was paralleled with an improvement in endothelial function as measured by flow-mediated dilation (FMD) [74]. The impairment of FMD in RA patients, was associated with age, IL-6, HDL, LDL and depleted EPCs population [75]. Of interest, despite similar levels of improvement in disease activity, the restoration of EPCs was attenuated in patients with higher bone-erosion scores than in those with lower scores. In RA patients with moderate disease activity, vitamin D deficiency is associated with a reduction in circulating levels of EPCs, suggesting that vitamin D might contribute to endothelial homeostasis in patients with RA [76]. 

All studies reporting data related to EPCs modification in RA are summarized in Table 1. An article search was performed on MEDLINE/PubMed using combinations of the following terms: ‘‘endothelial progenitor cells’’ and ‘‘rheumatoid arthritis”.

## 6. EPCs as a Biomarker for CV Comorbidities Related to RA

RA is characterized by a two-fold increase in the development of cardiovascular diseases (CVD), and CVD mortality is increased by approximately 50% in RA patients compared to the general population [77,78].

The relationship between EPCs and cardiovascular risk factors has been well described [79]. An impaired migratory response and negative correlation between EPCs and the severity of coronary artery disease has been described, and a reduced number of EPCs were observed in diabetic patients with peripheral artery disease [80,81]. Furthermore, the delivery of EPCs promoted the neovascularization of hindlimb ischemia and the direct myocardial injection of EPCs improved cardiac remodeling in different experimental models of myocardial ischemia [82,83]. Different studies have reported the association between EPC number, endothelial dysfunction and the enhancement of the atherosclerosis process in RA patients [65,69,70,75]. Indeed, coronary atherosclerosis was more prevalent in patients with RA compared to controls, and a multiple regression analysis revealed that older age (OR 1.25, 95% CI 1.10–1.41, *p* < 0.01) and lower EPCs (OR 0.07, 95% CI 0.00–0.97, *p* < 0.01) were independent predictors for coronary atherosclerosis in patients with RA [65]. Data from a recent study revealed that 60 months of preceding cumulative rheumatic inflammation was associated with altered osteocalcin expression in EPCs and acted as an increased risk of coronary calcification, suggesting that modulation of the bone-vascular axis by inflammation may play an important role in coronary calcification among RA patients [84]. Vascular calcification has been inversely correlated with bone mineral density, and low bone mass density appears to independently predict significant coronary artery disease in a population of predominantly women [85]. An evaluation of bone microarchitecture using a trabecular bone score, provided additional information regarding identification of RA patients at risk of the development of fractures [86], and the evaluation of a total-bone score in RA patients treated with anti-TNF allows for a greater discrimination of the population at lumbar spine fracture risk [87]. Furthermore, the reduction in the trabecular bone score for chronic inflammatory and autoimmune diseases was lower in patients with altered microvascular, as evaluated by nail video-capillaroscopy [3]. Interestingly, a recent study showed that elevated level of osteogenic circulating EPCs was associated with significantly higher risk of cardiac conduction abnormalities in subjects with RA [88].

Furthermore, a significantly higher EPC level was found in interstitial lung disease (ILD) patients affected by RA compared to RA patients without ILD, suggesting that an EPC increase may represent a reparative compensatory mechanism in patients with both RA and ILD [89].

It should be mentioned that across most of studies focused on EPC level and RA, the association between endothelial dysfunction and EPCs was present even though patients with a diagnosis of traditional CV risk factors such as hypertension, diabetes, smoking, dyslipidemia were excluded. It is reasonable to believe that RA-specific features, rather than traditional CV risk factors, trigger the modulation of EPCs, endothelial dysfunction and, finally, CV consequences and comorbidities.

## 7. RA Therapy and Modulation of EPCs Level and Function

Current RA treatment includes nonsteroidal anti-inflammatory drugs, glucocorticoids, synthetic and biological DMARDs [90]. Herbrig et al. [52] described that methotrexate, a synthetic DMARD, induced apoptosis in EPCs isolated from healthy controls and suggested that the reduction of EPCs observed in RA patients might in part be explained by methotrexate treatment. Ablin and colleagues [62] showed that after a single infusion of infliximab, (biological DMARD with anti-TNF action), in active seropositive RA patients, the level of EPCs increased significantly, by 33.4%, and EPCs adhesion and differentiation were also increased by 60% and 37.6%, respectively. Short term treatment with other subcutaneous biological DMARDs such as etanercept or adalimumab, increased EPCs level after three months [73].

Daily treatment with 25–50 mg of prednisolone for one week showed that the EPCs population significantly increased [53]. Experimental data have suggested that peroxisome proliferator-activated receptors *α* agonists, or fibrates, are important for EPCs differentiation; however, fenofibrate treatment for three months in twenty-seven patients with active RA and taking traditional disease-modifying antirheumatic drugs significantly improved the inflammatory and lipid profile, but failed to modify the number of EPCs. Preliminary data show that tocilizumab and hydroxychloroquine increase EPCs regeneration and differentiation in RA [91,92].

## 8. Conclusions and Future Approaches

The recruitment of EPCs and modification of their migratory and proliferation properties are crucial steps related to endothelial activation and dysfunction in RA. Multiple mechanisms, such as the activation of the inflammatory response, the generation of ROS and modulation of NO/eNOS signaling influence the functioning of EPCs and interfere with endothelial repair/damage equilibrium. Growing evidence from clinical studies suggests that RA is associated with a reduction in EPCs, and the modification of EPCs function. In addition, longer disease duration, activity status and treatments seem to influence their expression. It should be mentioned that the modification of EPCs has been observed in RA patients without cardiovascular risk factors, suggesting that the relationship between EPCs and RA may be influenced by RA-specific characteristics. Adiponectin, sphyngosine-1 phosphate signaling, and the modulation of the bone-vascular axis should be further studied as potential mechanisms involved in EPCs-RA relationship. Furthermore, EPCs levels seems to be associated with extra-articular complications and consequences of RA, such as atherosclerosis, coronary artery disease and interstitial lung disease. The functional modification of EPCs in RA disease appears to be a promising biomarker related to pathological RA progression, and future studies should better explore and clarify EPCs evolvement in these conditions. Further studies are required to investigate whether specific interventions that influence the role of EPCs in the preservation of the endothelial function in RA might serve as novel therapeutic strategies.

## Figures and Tables

**Figure 1 ijms-22-13675-f001:**
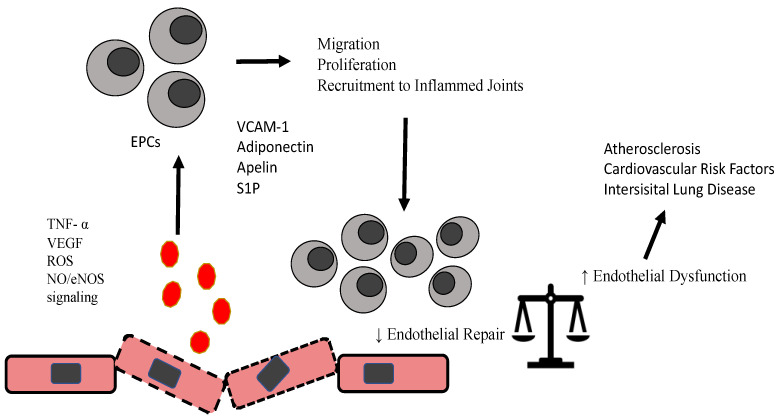
Schematic diagram highlighting modification of EPCs in RA.

**Table 1 ijms-22-13675-t001:** Endothelial Progenitor Cells in Rheumatoid Arthritis Patients.

First Author and Year	Study Population	Exclusion Criteria	Disease Activity State	MACs/ECFCs Identification	MACs/ECFCs Impairment in RA	Other Findings
Grisar J et al., 2005	52: 16 HC	Significant Hypertension, DM	active, low and no disease activity	CD34^+^/KDR^+^/AC133^+^	↓ MACs/ECFCs	MACs/ECFCs levels were inversely related to disease activity
Herbrig K et al., 2006	13 RA: 13 HC	DM, CAD, smokers.	low activity	CD34^+^/CD133^+^/KDR^+^	↓ MACs/ECFCs	Migratory activity of EPCs was reduced for RA patients. Adhesion to mature endothelial cells after activation with TNF-*α* was enhanced only in HC
Ablin J et al., 2006	14 RA receiving infliximab	DM, CAD, CVD, Claudication	active disease	CD31^+^/Tie-2	↑ ECFCs (after infliximab therapy)	Significant correlation was observed between the extent of clinical improvement and the level of increase in the number and function of EPCs
Grisar J et al., 2007	29 RA receiving GC	Significant Hypertension, DM, CVD, CAD.	moderate-high active disease	CD34^+^/KDR^+^/AC133^+^	↑ MACs/ECFCs (after GC therapy)	Disease activity and TNF decreased significantly after GC treatment.
Surdacki A et al., 2007	30 RA: 20 HC	Atherosclerosis, CV risk factors and Renal dysfunction. D.A.S < 3.2	active disease	CD34^+^/KDR^+^	↓ MACs	Plasma asymmetric dimethyl-L-arginine levels are ↑ in RA patients free of CV risk factors or disease
Egan C et al., 2008	36 RA: 30 HC	Acute macro- or microvascular events, DM, statin treatment	Moderate-high active disease	CD34^+^/CD133^+^ CD34^+^/CD117^+^ CD34^+^/CD31^+^ CD34^+^/KDR^+^ CD34^+^/CD133^+^/KDR^+^	MACs/ECFCs no significant difference	Levels of EPCs were negatively associated with prognostic markers of poor disease status
Kai-Hang Y et al., 2010	70 RA	CAD, myocardial infarction, stroke	N/R	CD34^+^ CD34^+^/KDR^+^ CD133^+^ CD133^+^/KDR^+^	↓ MACs	MACs predicted atherosclerosis in RA patients
Jodon de Villeroche V et al., 2010	59 RA: 36 HC	CV risk factors	different active disease	Lin^−^/7AAD^−^ CD34^+^/CD133^+^/VEGFR-2^+^	↑ ECFCs	No association between the EPCs and serum markers of inflammation or endothelial injury or synovitis.
Rodriguez- Carrio J et al., 2012	83 RA: 13 HC	CV risk factors	early stage	CD34^+^/VEGFR2^+^/CD133^+^	MACs unchanged	EPCs number exhibited a positive correlation with disease activity in early RA
Shirinsky I et al., 2012	25 RA: 16 Osteoarthritis	N/R	active disease	CD34+/CD144+/CD3^−^	↓ MACs/ECFCs	After 12 weeks of treatment with fenofibrate, no significant changes were observed in EPCs levels
Spinelli F.R et al., 2013	17 RA: 12 HC	CVD, CKD; Dyslipidemia, DM	active state	CD34^+^/KDR^+^	↓ MACs	Short-term treatment with anti-TNF was able to increase circulating EPCs
Rodriguez- Carrio J. et al., 2014	120 RA: 52 HC	N/R	different active disease	CD34^+^/VEGFR2^+^/CD133^+^	↓ MACs	EPCs reduced in patients with low IFNα
Rodriguez- Carrio J. et al., 2015	103 RA: 18 HC	N/R	different active disease	CD34^+^/VEGFR2^+^/CD133^+^	↓ MACs	Angiogenic T cells are reduced in RA and are associated with CV risk factors
Park YJ et al., 2014	126 RA: 26 HC	CAD, stroke, CKD, CHF	different active disease	CD34^+^/VEGFR-2^+^	↓ MACs	EPCs is independently associated with bone erosion scores in RA patients. serum CXCL12 level is significantly higher in RA patients.
Rodriguez- Carrio J et al., 2014	194 RA	N/R	different active disease	CD34^+^/VEGFR2^+^/CD133^+^	↓ MACs (>1 year RA vs. <, =1 year)	RDW was associated with ↓ EPCs and increased levels of different mediators linked to endothelial damage
Lo Gullo A et al., 2015	27 RA: 41 HC	CV risk factors, Vitamin D treatment	moderate disease activity	CD34^+^	↓ MACs	Vitamin D deficiency is associated with ↓ MACs
Verma I et al, 2015	35 RA: 25 HC	CV risk factors	N/R	CD34^+^/CD133^+^	↓ MACs	Age, IL-6, HDL, LDL and ↓ EPCs predicted accelerated atherosclerosis

RA: Rheumatoid Arthritis; HC: healthy Controls; DM: Diabetes Mellitus; CAD: Coronary Artery Disease; CVD: Cerebrovascular Disease; GC: Glucocorticoid; D.A.S: Disease Activity Score; N/A: Not applicable or not specified data; CKD: Chronic Kidney DIsease; CHF: Chronic Heart Failure; KDR: Kinase Insert Domain Receptor; MACs: Myeloid Angiogenic Cells; ECFCs: Endothelial Colony Forming Cells; IFN-*α*: Interferon alfa; VEGF: Vascular Endothelial Growth Factor. ↓: reduced; ↑: increased.

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
