# Peer review of "Endothelial Progenitor Cells and Rheumatoid Arthritis: Response to Endothelial Dysfunction and Clinical Evidences"

_ijms, 2021, doi:10.3390/ijms222413675_

Round 1
Reviewer 1 Report
1) L30-36. Introduction. Rheumatoid Arthritis (RA) is a chronic inflammatory joint disease, with a prevalence of about 1 % worldwide. The peak incidence is registered among subjects of 60 years of age, and women are characterized by a higher risk of developing RA. Pain, swelling and
stiffness of feet and hands joints is the classic presentation of the diseases. However, non-specific symptoms as fatigue and fever may appear before classic symptoms. Extra-articular manifestations of RA include: nodules, rheumatoid vasculitis, progressive disability and lower quality of life. Interstitial lung disease, neurological, atherosclerosis and cardiovascular disease are common comorbidities associated with RA. Please improve this paragraph and add these sentences:
a- Ghazi, M., Kolta, S., Briot, K., Fechtenbaum, J., Paternotte, S., & Roux, C. (2012). Prevalence of vertebral fractures in patients with rheumatoid arthritis: revisiting the role of glucocorticoids. Osteoporosis international : a journal established as result of cooperation between the European Foundation for Osteoporosis and the National Osteoporosis Foundation of the USA, 23(2), 581–587. https://doi.org/10.1007/s00198-011-1584-3
b- Ruaro, B., Casabella, A., Paolino, S., Pizzorni, C., Alessandri, E., Seriolo, C., Botticella, G., Molfetta, L., Odetti, P., Smith, V., & Cutolo, M. (2018). Correlation between bone quality and microvascular damage in systemic sclerosis patients. Rheumatology (Oxford, England), 57(9), 1548–1554. https://doi.org/10.1093/rheumatology/key130
2) Introduction. L 46-55. Activation of inflammatory response is characterized by infiltration of T and B cells, monocytes, endothelial cells (ECs), fibroblasts proliferation, and autoimmune process further amplifies
this process. In addition, activation of pro-inflammatory cytokines, as tumor necrosis factor (TNF) and IL-6, mediate the release of prostaglandins and metalloproteinases (MMP), which in turn induce symptoms and signs development. The sustained immune response and chronic inflammation, which characterize RA, may induce endothelial activation,
damage and dysfunction. An equilibrium between endothelial damage and repair, together with preservation of endothelial integrity is of crucial importance for the homeostasis of the endothelium. Endothelial Progenitor Cells (EPCs) represent a heterogenous cell population, characterized by the ability to differentiate into mature (ECs) which contribute to vascular homeostasis, neovascularization and endothelial repair. Please improve this section and add some references:
a- Aspal, M., & Zemans, R. L. (2020). Mechanisms of ATII-to-ATI Cell Differentiation during Lung Regeneration. International journal of molecular sciences, 21(9), 3188. https://doi.org/10.3390/ijms21093188
b- Ruaro, B., Salton, F., Braga, L., Wade, B., Confalonieri, P., Volpe, M. C., Baratella, E., Maiocchi, S., & Confalonieri, M. (2021). The History and Mystery of Alveolar Epithelial Type II Cells: Focus on Their Physiologic and Pathologic Role in Lung. International journal of molecular sciences, 22(5), 2566. https://doi.org/10.3390/ijms22052566
3) Introduction. L 61-66. However, given the crucial connection between
EPCs, endothelial damage and chronic inflammation, a modification of EPC function should be present in RA. In this review, we discuss about EPC role in endothelial homoeostasis, describe the relevance of endothelial dysfunction in RA, illustrate the evidence regarding EPC modification in RA, and EPC role as a biomarker in cardiovascular comorbidities related to RA. Please ameliorate the description of study aim.
4) Please improve the links between paragraphs, they seem a bit unrelated
5) L 260-266. Data from a recent study revealed that preceding 60 months of cumulative rheumatic inflammation was associated with altered
osteocalcin expression in EPCs and increased risk of coronary calcification, suggesting that modulation of the bone–vascular axis by inflammation may play an important role in coronary calcification among RA patients 59. Of note a recent study showed that elevated level of osteogenic circulating EPCs was associated with significantly higher risk of cardiac conduction abnormalities in subjects with RA. Please add some references as reported in comment 1.
6) Conclusions and Future Approaches L 277-284
Inflammatory response, which characterizes RA, is able to modify the recruitment, migratory and proliferation properties of EPCs. Multiple mechanisms are involved and the majority of clinical data in RA patients reported a reduction in EPC number, including patients with RA compared to healthy controls. Furthermore, longer disease duration, activity status and treatments seem to influence EPC expression. Functional modification of
EPCs in RA disease appears as a promising biomarker related to pathological RA progression and future studies should explore and clarify better EPC evolvement in these conditions. Please improve the conclusions and underline the novelty of your article.
Author Response
Reviewer 1
1) L30-36. Introduction. Rheumatoid Arthritis (RA) is a chronic inflammatory joint disease, with a prevalence of about 1 % worldwide. The peak incidence is registered among subjects of 60 years of age, and women are characterized by a higher risk of developing RA. Pain, swelling and stiffness of feet and hands joints is the classic presentation of the diseases. However, non-specific symptoms as fatigue and fever may appear before classic symptoms. Extra-articular manifestations of RA include: nodules, rheumatoid vasculitis, progressive disability and lower quality of life. Interstitial lung disease, neurological, atherosclerosis and cardiovascular disease are common comorbidities associated with RA. Please improve this paragraph and add these sentences:
a- Ghazi, M., Kolta, S., Briot, K., Fechtenbaum, J., Paternotte, S., & Roux, C. (2012). Prevalence of vertebral fractures in patients with rheumatoid arthritis: revisiting the role of glucocorticoids. Osteoporosis international : a journal established as result of cooperation between the European Foundation for Osteoporosis and the National Osteoporosis Foundation of the USA, 23(2), 581–587. https://doi.org/10.1007/s00198-011-1584-3
b- Ruaro, B., Casabella, A., Paolino, S., Pizzorni, C., Alessandri, E., Seriolo, C., Botticella, G., Molfetta, L., Odetti, P., Smith, V., & Cutolo, M. (2018). Correlation between bone quality and microvascular damage in systemic sclerosis patients. Rheumatology (Oxford, England), 57(9), 1548–1554. https://doi.org/10.1093/rheumatology/key130
Reply:Thank You for the useful suggestion. We modified the text in L30-36, and added a description of other extra-articular and systemic complications related to RA with references. In the revised version of our manuscript in lines 34-41 you can find the following text:
''Extra-articular manifestations and complications of RA include: nodules, rheumatoid vasculitis, osteoporosis, vertebral fractures, progressive disability and lower quality of life. Compared to healthy controls, RA patients may have lower bone quality and RA is considered a risk factor for vertebral fractures. Patients with severe RA disease have a 6.5- fold higher risk for vertebral fractures development, compared to age-matched controls and current use of steroids and disease-modifying anti-rheumatic drugs are inversely related to vertebral fractures''.
2) Introduction. L 46-55. Activation of inflammatory response is characterized by infiltration of T and B cells, monocytes, endothelial cells (ECs), fibroblasts proliferation, and autoimmune process further amplifies this process. In addition, activation of pro-inflammatory cytokines, as tumor necrosis factor (TNF) and IL-6, mediate the release of prostaglandins and metalloproteinases (MMP), which in turn induce symptoms and signs development. The sustained immune response and chronic inflammation, which characterize RA, may induce endothelial activation, damage and dysfunction. An equilibrium between endothelial damage and repair, together with preservation of endothelial integrity is of crucial importance for the homeostasis of the endothelium. Endothelial Progenitor Cells (EPCs) represent a heterogenous cell population, characterized by the ability to differentiate into mature (ECs) which contribute to vascular homeostasis, neovascularization and endothelial repair. Please improve this section and add some references:
a- Aspal, M., & Zemans, R. L. (2020). Mechanisms of ATII-to-ATI Cell Differentiation during Lung Regeneration. International journal of molecular sciences, 21(9), 3188. https://doi.org/10.3390/ijms21093188
b- Ruaro, B., Salton, F., Braga, L., Wade, B., Confalonieri, P., Volpe, M. C., Baratella, E., Maiocchi, S., & Confalonieri, M. (2021). The History and Mystery of Alveolar Epithelial Type II Cells: Focus on Their Physiologic and Pathologic Role in Lung. International journal of molecular sciences, 22(5), 2566. https://doi.org/10.3390/ijms22052566
Reply: We introduced a more detailed description of the mechanisms involved in the pathogenesis of RA, the inflammatory response and consequences of immune tolerance failure on fibroblasts and alveolar epithelial cells activation and transformation. Please find in the revised version of our manuscript lines 63-92 the following text:
''The onset of RA is proceeded by a preclinical RA stage, where genetic factors like shared-epitope-positive HLA-DRB1 alleles, PTPN22 variant and environmental stressors promote post-translational protein modifications such as citrullination or carbamylation, and generate neo-epitopes of autologous proteins like fibrinogen, fibronectin, collagen, and vimentin. This process results in loss of self-tolerance and development of autoan-tibodies against anticitrullinated protein antibodies, antibodies directed against the Fc portion of immunoglobulins, and RF. Consequently immune response results enlarged by activation and differentiation of T cells, release of IL-2, Il-6,Il-17, Il-21, INF-gamma and activation of B-cells which secret autoantibodies . Abnormal differentiation of naive CD4+ T cells into highly proliferative and proinflammatory effector cells, leads to tissue tolerance failure and early synovitis. Defective transition of T and B cells population from naive to effector and memory states has been suggested as the principal mechanism involved in the development of tissue tolerance. Of note, in experimental models of arthritis, in vitro generated collagen-II specific B cell induced immune tolerance 14. Il-17, INF- γ and immune complexes activate macrophages, which release IL-1, IL-6, TNF and activate fibroblasts. Fibroblasts activation and proliferation may adopt pro-inflammatory and tissue invasive functions by release of metalloproteinases (MMPs) and differentiations of macrophages to osteoclasts. In addition, release of IL-11 by activated fibroblasts, controls fibroblasts trafficking and production of IL-8 and VEGF which results in angiogenesis and potentiates neovascularization15. Evasion of synovial membrane by macrophages and fibroblasts leads to cartilage degradation and bone erosion. The sustained immune response and chronic inflammation, which characterize RA, may induce endothelial activation, damage and dysfunction. Furthermore, epithelial- mesenchymal transition has been associated to RA and IL-23 seems to intermediate the transition of alveolar epithelial cells (ATI) to a mesenchymal phenotype through mTOR/S6 signaling. ATII cells have an important role in the regulation of innate immunity and inflammatory mediators as transforming growth factor β, increased in RA, may modify ATII secretory profile which in turn may directly activate fibroblasts migration and proliferation resulting in lung tissue remodeling and fibrosis development''.
3) Introduction. L 61-66. However, given the crucial connection between
EPCs, endothelial damage and chronic inflammation, a modification of EPC function should be present in RA. In this review, we discuss about EPC role in endothelial homoeostasis, describe the relevance of endothelial dysfunction in RA, illustrate the evidence regarding EPC modification in RA, and EPC role as a biomarker in cardiovascular comorbidities related to RA. Please ameliorate the description of study aim.
Reply: Study aim was modified as follows (lines 103-106 in abstract as well):
''In this review we describe EPCs response to endothelial modification in RA with the aim to illustrate current evidences regarding EPCs level and function in RA, summarize EPCs role as a biomarker in cardiovascular comorbidities related to RA and finally discuss modulation of EPCs secondary to RA therapy.''
4) Please improve the links between paragraphs, they seem a bit unrelated
Reply: To improve the links between paragraphs the following sentences were added:
for paragraph 2 and 3 lines 160-165:
''Impairment of endothelial function has been established as a key element in the development of atherosclerosis process and is recognized as an important factor related to cardiovascular risk in RA. Likewise, altered endothelial reactivity has been documented in RA patients without cardiovascular risk factors and prior to atherosclerotic plaque detection, suggesting that endothelial impairment is related to a specific RA associated chronic inflammatory condition''.
for paragraph 3 and 4 lines 221-224:
''Maintaining adequate levels and function of EPCs is of particular importance for the preservation of endothelial function and protective against atherosclerosis process, as the chronic inflammatory condition which characterizes RA jeopardizes the endothelial integrity''.
and for paragraph 5 and 6 lines 360-364.
''Impaired migratory response and negative correlation between EPCs and severity of coronary artery disease has been described and reduced number of EPCs were observed in diabetic patients with peripheral artery disease. Furthermore, EPCs delivery promoted neovascularization of hindlimb ischemia and direct myocardial injection of EPCs improved cardiac remodeling in different experimental models of myocardial ischemia''.
5) L 260-266. Data from a recent study revealed that preceding 60 months of cumulative rheumatic inflammation was associated with altered
osteocalcin expression in EPCs and increased risk of coronary calcification, suggesting that modulation of the bone–vascular axis by inflammation may play an important role in coronary calcification among RA patients 59. Of note a recent study showed that elevated level of osteogenic circulating EPCs was associated with significantly higher risk of cardiac conduction abnormalities in subjects with RA. Please add some references as reported in comment 1.
Reply: We extended this part and added the following sentences lines 381-390 :
''Vascular calcification has been inversely correlated with bone mineral density low bone mass density appears to independently predict significant coronary artery disease in a population of predominantly women. Evaluation of bone microarchitecture by trabecular bone score, has provided additional information regarding identification of RA patients at risk for fractures development, and evaluation of total bone score in RA patients on treatment with anti-TNF allows for a greater discrimination of the population at lumbar spine fracture risk. Furthermore, reduction of trabecular bone score present in chronic inflammatory and autoimmune diseases was lower in patients with altered microvascular as evaluated by nail video-capillaroscopy.''
6) Conclusions and Future Approaches L 277-284
Inflammatory response, which characterizes RA, is able to modify the recruitment, migratory and proliferation properties of EPCs. Multiple mechanisms are involved and the majority of clinical data in RA patients reported a reduction in EPC number, including patients with RA compared to healthy controls. Furthermore, longer disease duration, activity status and treatments seem to influence EPC expression. Functional modification ofEPCs in RA disease appears as a promising biomarker related to pathological RA progression and future studies should explore and clarify better EPC evolvement in these conditions. Please improve the conclusions and underline the novelty of your article.
Reply:
We extended the conclusions and future approach section and underlined the novelty of our review. Please find the following text in the revised version of our manuscript:
''Recruitment of EPCs and modification of their migratory and proliferation properties is a crucial step related to endothelial activation and dysfunction in RA. Multiple mechanisms as activation of inflammatory response, generation of ROS and modulation of NO/eNOS signaling influence EPCs functioning and interfere with endothelial repair / damage equilibrium. Growing evidence from clinical studies suggests that RA is associated with reduction of EPCs numbers, modification EPCs function. In addition, longer disease duration, activity status and treatments seem to influence their expression. It should be mentioned that modification of EPCs is observed in RA patients without cardiovascular risk factors, suggesting that the relationship between EPCs and RA may be influenced by RA specific characteristics. Adiponectin, sphyngosine-1 phosphate signaling and modulation of bone-vascular axis should be further studied as potential mechanisms involved in EPCs-RA relationship. Furthermore, EPCs level seems to be associated with extra-articular complications and consequences of RA as atherosclerosis, coronary artery disease and interstitial lung disease. Functional modification of EPCs in RA disease appears as a promising biomarker related to pathological RA progression and future studies should explore and clarify better EPCs evolvement in these conditions. Further studies are required to investigate whether specific interventions that influence EPCs role on the preservation of endothelial function in RA might serve as novel ther-apeutic strategies''.
We wish to thank the Reviewer for the constructive criticism, suggestions and comments which helped us to significantly improve the quality of our study.
Klara Komici MD,
University of Molise
Department of Medicine and Health Sciences,
Via Giovanni Paolo II,86100 Campobasso, Italy
phone:00390874404710
fax: 00390874404710
email: klara.komici@unimol.it
Reviewer 2 Report
- The review summarises the endothelial progenitor cell (EPC) response to rheumatoid arthritis (RA) and related mechanisms. It would be better if a specific section elaborates the prospect of EPC as a target in terms of RA therapy based on current evidence.
- There are grammatical errors and stilted use of language in the manuscript. English language correction throughout the manuscript by a native speaker is needed before it is published.
- Check all abbreviations for any words to ensure that only once do not require abbreviations. Also, more attention should be paid to the format and standardization of the reference descriptions.
- The past and present tense is used interchangeably throughout the manuscript. Please use present tense whenever possible (Example, line 184: use ….. is associated with …. instead of …… was associated with …..)
- Check the use of “EPC number” and “EPC numbers” throughout the manuscript. Often the singular form is used when it should be plural.
- There is no citation for Tab. 1 and Fig. 1 in the text.
- It will benefit to broaden include a section on how blood vessels and their pericytes, in general, contribute to RA, such as pericyte to fibroblast differentiation PMID: 33536212
Author Response
Reviewer 2
The review summarises the endothelial progenitor cell (EPC) response to rheumatoid arthritis (RA) and related mechanisms. It would be better if a specific section elaborates the prospect of EPC as a target in terms of RA therapy based on current evidence.
Reply: Thank you for the suggestion. We added a paragraph regarding RA therapy and modification of EPCs, where we summarized the current evidence. In the revised version of our manuscript you can find the following text:
''Current RA treatment includes nonsteroidal anti-inflammatory drugs, glucocorti-coids, synthetic and biological DMARDs. Herbrig at al described that methotrexate a synthetic DMARD, induced apoptosis in EPCs isolated from healthy controls and suggested that in part the reduction of EPCs number observed in RA patients in part might be explained by methotrexate treatment. Ablin and colleagues showed that after a single infusion of infliximab, (biological DMARD with anti-TNF action) , in active seropositive RA patients, EPCs level increased significantly by 33.4% and EPCs adhesion and differentiation were also increased respectively by 60% and 37.6 %. Short term treatment with other subcutaneous biological DMARDs as etanercept or adalimumab, increased EPCs level after three months.
Daily treatment with 25-50 mg of prednisolone for one week showed that EPCs population significantly increased. Experimental data have suggested that peroxisome proliferator-activated receptors α agonists, or fibrates, are important for EPCs differentiation, however, fenofibrate treatment for three months in twenty-seven patients with active RA taking traditional disease-modifying antirheumatic drugs significantly im-proved the inflammatory and lipid profile but failed to modify EPCs numbers. Pre-liminary data show that tocilizumab and hydroxychloroquine increase EPCs regeneration and differentiation in RA.''
There are grammatical errors and stilted use of language in the manuscript. English language correction throughout the manuscript by a native speaker is needed before it is published.
Check all abbreviations for any words to ensure that only once do not require abbreviations. Also, more attention should be paid to the format and standardization of the reference descriptions.
Reply: We checked the text. Unnecessary abbreviations were canceled and the necessary ones were introduced.
The past and present tense is used interchangeably throughout the manuscript. Please use present tense whenever possible (Example, line 184: use ….. is associated with …. instead of …… was associated with …..)
Reply: Thank You! this was corrected and also in the text where was possible we introduced present tense.
Check the use of “EPC number” and “EPC numbers” throughout the manuscript. Often the singular form is used when it should be plural.
Reply:Number was modified in numbers, where the plural form was necessary.
There is no citation for Tab. 1 and Fig. 1 in the text.
Reply: We introduced in the text citation for both table 1 and figure 1. Please check lines 277 and 339.
It will benefit to broaden include a section on how blood vessels and their pericytes, in general, contribute to RA, such as pericyte to fibroblast differentiation PMID:
33536212
Reply:Thank you for the suggestions. We included a paragraph in the 4th section where we discussed about the vascularization process in RA, and underlined the role of immature vascularization, VEGF, perycytes and their differentiation toward fibroblasts.
In the revised version of our manuscript you can find the following text:
''In RA combination of local hypoxic conditions and release of inflammatory cytokines as TNF, IL-1, IL-6 and IL-18 activate macrophages and synovial tissue fibroblasts to secrete VEGF and fibroblasts growth factor. In turn they activate ECs, induce the production of pro- proteolytic enzymes and basement membrane degradation by MMPs, which results in ECs migration and proliferation to vascular tubules, and lastly pericytes are incorporated into the newly formed basement membrane. Excessive expression of VEGF in RA inflamed synovial tissue has been broadly reported and double labeling of endothelium and pericytes/smooth muscle mural cells of synovial arthroscopic biopsies from RA, revealed that immature vessels were present since the earliest phases of RA and their density was increased in patients with longer disease duration. Of interest a recent study observed that pericyte-derived fibroblasts contribute to fibroblast proliferation and fibrosis expansion in arthritis''.
We wish to thank the Reviewer for the constructive criticism, suggestions and comments which helped us to significantly improve the quality of our study.
Klara Komici MD,
University of Molise
Department of Medicine and Health Sciences,
Via Giovanni Paolo II,86100 Campobasso, Italy
phone:00390874404710
Reviewer 3 Report
This is a concise review about EPC role in endothelial homoeostasis and endothelial dysfunction in RA. The table and graph are helpful. The review is lacking any mention of the role of immune tolerance as a potential route in amelioration of RA (PMID: 33921248). It does not discuss the role of IL-11 either (PMID: 29327326). Both immune tolerance and IL-11 roles need to be discussed with appropriate references.
Author Response
Reviewer 3
This is a concise review about EPC role in endothelial homoeostasis and endothelial dysfunction in RA. The table and graph are helpful. The review is lacking any mention of the role of immune tolerance as a potential route in amelioration of RA (PMID: 33921248). It does not discuss the role of IL-11 either (PMID: 29327326). Both immune tolerance and IL-11 roles need to be discussed with appropriate references.
Reply:Thank you for the review of our article and your helpful comments. As suggested by the Reviewer we added a paragraph in introduction section, where we discussed the role of immune tolerance in RA and the role of IL-11. In lines 63-84 of the revised version of the manuscript you can find the above text with references:
''The onset of RA is proceeded by a preclinical RA stage, where genetic factors like shared-epitope-positive HLA-DRB1 alleles, PTPN22 variant and environmental stressors promote post-translational protein modifications such as citrullination or carbamylation, and generate neo-epitopes of autologous proteins like fibrinogen, fibronectin, collagen, and vimentin. This process results in loss of self-tolerance and development of autoantibodies against anticitrullinated protein antibodies, antibodies directed against the Fc portion of immunoglobulins, and RF. Consequently immune response results enlarged by activation and differentiation of T cells, release of IL-2, Il-6, Il-17, Il-21, INF-gamma and activation of B-cells which secret autoantibodies . Abnormal differentiation of naive CD4+ T cells into highly proliferative and proinflammatory effector cells, leads to tissue tolerance failure and early synovitis. Defective transition of T and B cells population from naive to effector and memory states has been suggested as the principal mechanism involved in the development of tissue tolerance. Of note, in experimental models of arthritis, in vitro generated collagen-II specific B cell induced immune tolerance 14. Il-17, INF- γ and immune complexes activate macrophages, which release IL-1, IL-6, TNF and activate fibroblasts. Fibroblasts activation and proliferation may adopt pro-inflammatory and tissue invasive functions by release of metalloproteinases (MMPs) and differentiations of macrophages to osteoclasts. In addition, release of IL-11 by activated fibroblasts, controls fibroblasts trafficking and production of IL-8 and VEGF which results in angiogenesis and potentiates neovascularization. Evasion of synovial membrane by macrophages and fibroblasts leads to cartilage degradation and bone erosion.''
We wish to thank the Reviewer for the constructive criticism, suggestions and comments which helped us to significantly improve the quality of our study.
Klara Komici MD,
University of Molise
Department of Medicine and Health Sciences,
Via Giovanni Paolo II,86100 Campobasso, Italy
phone:00390874404710
fax: 00390874404710
Round 2
Reviewer 2 Report
The authors have addressed all my comments and suggestions. I do not have any further comments.